# Incidence of Hereditary Gastric Cancer May Be Much Higher than Reported

**DOI:** 10.3390/cancers14246125

**Published:** 2022-12-12

**Authors:** Paula Baraúna de Assumpção, Paulo Pimentel de Assumpção, Fabiano Cordeiro Moreira, Ândrea Ribeiro-dos-Santos, Amanda F. Vidal, Leandro Magalhães, André Salim Khayat, André Maurício Ribeiro-dos-Santos, Giovanna C. Cavalcante, Adenilson Leão Pereira, Inácio Medeiros, Sandro José de Souza, Rommel Mario Rodríguez Burbano, Jorge Estefano Santana de Souza, Sidney Emanuel Batista Dos Santos

**Affiliations:** 1Oncology Research Center, Federal University of Pará, Belém 66073-005, Pará, Brazil; 2Laboratory of Human and Medical Genetics, Institute of Biological Sciences, Graduate Program of Genetics and Molecular Biology, Federal University of Pará, Belém 66075-110, Pará, Brazil; 3Bioinformatics Department, Federal University of Rio Grande do Norte, Natal 59078-400, Rio Grande do Norte, Brazil

**Keywords:** hereditary gastric cancer, germline mutation, epidemiology, exome analyses, ancestry

## Abstract

**Simple Summary:**

Hereditary gastric cancer is supposed to be a rare condition and depends on the presence of a recognized pathogenic mutation. Nonetheless, many cases fulfill the clinical criteria for HGC but lack those mutations. This report suggests new mutations to be included as putatively involved in HGC risk. A hypothesis of genetic ancestry influencing the identification of HGC mutations is sustained, since never-reported mutations were revealed to occur in Amerindians and, mainly, in admixed populations with a significant Amerindians genetic background, suffering early-onset gastric cancer. These populations are under-represented in international data banks of cancer mutations, and these newly reported mutations may pave the way for incorporating additional lab tests to discover HGC, favoring early diagnosis and risk-reduction procedures, improving clinical outcomes, and contributing to HGC control.

**Abstract:**

Hereditary gastric cancers (HGCs) are supposed to be rare and difficult to identify. Nonetheless, many cases of young patients with gastric cancer (GC) fulfill the clinical criteria for considering this diagnosis but do not present the defined pathogenic mutations necessary to meet a formal diagnosis of HGC. Moreover, GC in young people is a challenging medical situation due to the usual aggressiveness of such cases and the potential risk for their relatives when related to a germline variant. Aiming to identify additional germline alterations that might contribute to the early onset of GC, a complete exome sequence of blood samples from 95 GC patients under 50 and 94 blood samples from non-cancer patients was performed and compared in this study. The number of identified germline mutations in GC patients was found to be much higher than that from individuals without a cancer diagnosis. Specifically, the number of high functional impact mutations, including those affecting genes involved in medical diseases, cancer hallmark genes, and DNA replication and repair processes, was much higher, strengthening the hypothesis of the potential causal role of such mutations in hereditary cancers. Conversely, classically related HGC mutations were not found and the number of mutations in genes in the CDH1 pathway was not found to be relevant among the young GC patients, reinforcing the hypothesis that existing alternative germline contributions favor the early onset of GC. The LILRB1 gene variants, absent in the world’s cancer datasets but present in high frequencies among the studied GC patients, may represent essential cancer variants specific to the Amerindian ancestry’s contributions. Identifying non-reported GC variants, potentially originating from under-studied populations, may pave the way for additional discoveries and translations to clinical interventions for GC management. The newly proposed approaches may reduce the discrepancy between clinically suspected and molecularly proven hereditary GC and shed light on similar inconsistencies among other cancer types. Additionally, the results of this study may support the development of new blood tests for evaluating cancer risk that can be used in clinical practice, helping physicians make decisions about strategies for surveillance and risk-reduction interventions.

## 1. Introduction

Hereditary gastric cancers (HGCs) are supposed to be rare and difficult to identify. Nonetheless, there are many cases of young patients with gastric cancer (GC) and many families with multiple cases of GC [1,2]. These situations are potentially related to the genetic backgrounds of patients, since most sporadic GC affects older patients and does not present familial aggregations [3,4].

Reaching a molecular diagnosis of HGC is an arduous task, even in referential cancer centers able to perform every type of molecular investigation. Most clinically considered cases fail to achieve the expected requirements to affirm a molecular diagnosis, regardless of being diagnosed as potential cases of HGC [5].

The consequences of such disparities affect clinical practice. Many physicians feel insecure about managing such patients due to concerns about performing screening procedures that might contribute to potential early diagnosis among patients’ relatives but may also result in stress and anxiety for them, in addition to being expensive and invasive. Another critical issue is that a late diagnosis in such cases could cause embarrassment for physicians and damage to patients [6,7].

Despite exponential increases in medical genetic knowledge and technical developments, many questions still need to be solved to improve the diagnosis of HGC [8]. Hereditary driver mutations constitute the rarest type of driver mutations and are usually not enough to cause cancer, since having every necessary mutation for an adult cancer occurrence already presented at birth seems to be inconceivable [9,10].

Conversely, starting the cancer process with driver mutations might bring an advantage, possibly anticipating cancer onset many years in advance [11]. If confirmed, this hypothesis could explain why most hereditary cancer cases occur among younger people, in comparison with non-hereditary ones.

Undoubtedly, more information is needed to explain diverse clinical situations, including cases where a germline contribution is strongly supposed but not recognized, according to the current rules [7]. This is the case in young gastric cancer patients without CDH1 germline cancer causal mutations or other rare ones, such as those in the CTNN1A gene, that are supposed to strongly favor GC onset [12,13,14,15].

Many potential clinical cases of HGC do not seem to harbor a pathogenic germline mutation. Nonetheless, germline mutations, including functional ones, in essential cancer-related genes are found through the exome sequencing of these patients, even though these are not recognized as causal events for hereditary gastric cancer [16,17].

Aiming to explore potential germline contributions to HGC, as well as for other hereditary cancer types, we proposed a strategy consisting of searching for germline mutations affecting potential cancer-related genes and not limited to the already established HGC-causing genes.

Additionally, we hypothesized that the global germline mutation burden and specific germline mutation burdens, such as those in DNA repair genes or genes involved in the DNA replication process (among others described in detail in the Methods session), might impact the risk for HGC.

This research strategy also included the investigation of under-studied populations from the Amazonia, since the scarcity of the sequencing of Amerindians’ genomes might represent a window of opportunities for discovering new germline variants involved in GC risk.

## 2. Methods

The study was approved by the National Research Ethics Committee (CONEP; available at: http://conselho.saude.gov.br/comissoes-cns/conep/, accessed on 13 January 2022) and by the Research Ethics Committee of the Federal University of Pará (CAE: 20654313.6.0000.5172). All participating individuals and community leaders signed an informed consent form (TCLE), and the research followed relevant guidelines/regulations.

Ninety-five patients with an established histopathological diagnosis of gastric adenocarcinoma under 50 years old were included. Patients who submitted to curative intended gastrectomy and had a peripheral blood sample collected before any therapeutic modality were selected for analysis. Those with a previous or concomitant diagnosis of other cancers, inadequate sample quality for sequencing, or inaccessibility of data in clinical records were excluded.

The clinical and pathological data of the patients are presented in Appendix A.

A non-cancer control group of 94 Amerindians from the Brazilian Amazonia was also included. The senior author (Santos, S.E.) provided data for this group that are available at the European Nucleotide Archive database (ENA https://www.ebi.ac.uk/ena, accessed on 9 September 2022) under accession number PRJEB35045.

The same lab protocols were applied for every step of the analyses of both groups.

### 2.1. Extraction of the DNA and Preparation of the Exome Library

DNA was extracted from a peripheral blood sample using the phenol–chloroform method described by Green M.R. and Sambrook J. [18]. A Nanodrop 8000 spectrophotometer (Thermo Fisher Scientific Inc., Wilmington, DE, USA) was used to quantify the genetic material. Its integrity was evaluated with 2% agarose gel electrophoresis.

Libraries were prepared using the Nextera Rapid Capture Exome (Illumina^®^, San Diego, CA, USA) and SureSelect Human All Exon V6 (Agilent Technologies, Santa Clara, CA, USA) kits, following the manufacturer’s recommendations. The sequencing reactions were performed on the NextSeq 500^®^ platform (Illumina^®^, San Diego, CA, USA) using the NextSeq 500 High-output v2 Kit 300 cycle kit (Illumina^®^, San Diego, CA, USA).

### 2.2. DryLab

Reads in FASTQ format were analyzed for quality (FastQC v.0.11 http://www.bioinformatics.babraham.ac.uk/projects/fastqc/, accessed on 9 September 2022) and filtered to eliminate low-quality reads (fastx_tools v.0.13 http://hannonlab.cshl.edu/fastx_toolkit/, accessed on 9 September 2022). Then, the sequences were aligned to the reference genome (GRCh38) using the BWA v.0.7 tool (http://bio-bwa.sourceforge.net/, accessed on 9 September 2022).

After alignment, the generated files were indexed and sorted (SAMtools v.1.2—http://sourceforge.net/projects/samtools/, accessed on 9 September 2022). Subsequently, the alignments were processed to remove duplicate readings (Picard Tools v.1.129—http://broadinstitute.github.io/picard/, accessed on 9 September 2022), recalibrate mapping quality, and enact local realignment (GATK v.3.2—https://www.broadinstitute.org/gatk/, accessed on 9 September 2022). Finally, the results were processed in search of variants (GATK v.3.2, USA; https://gatk.broadinstitute.org/hc/en-us, accessed on 9 September 2022) of the reference genome.

### 2.3. Statistical Analyses

All graphical and statistical analyses were performed using the R Studio v.4.2.1 program (R Foundation for Statistical Computing, Vienna, Austria). Significant differences in mutation rates were analyzed with the Wilcoxon–Mann–Whitney and Kruskal–Wallis tests. The false discovery rate (FDR) was used to correct all analyses. Results were considered statistically significant when the adjusted *p*-value was less than 0.01 (adj. *p* ≤ 0.01).

The impact and consequences of mutations were described with SnpEff software, a program for annotating and predicting the effects of single-nucleotide polymorphisms [19]. SIFT (Sorting Intolerant from Tolerant) and PolyPhen [20] bioinformatics tools were used to predict the functional effect of missense mutations.

To report the number of mutations for each subset of the genes described below, only non-synonymous mutations with a high or moderated impact and frequencies under 10% of the world population were considered. Regarding the total germline mutation burden (GMB), every identified mutation was included without applying these filters.

The gene subsets included in the analyses and applied for the group’s comparisons were as follows:
GMB: the total number of germline mutations.GHFI: the number of non-synonymous mutations presenting high functional impact, according to Qing et al. [21].GMed: the number of non-synonymous mutations presenting high functional impact among genes related to clinical diseases [21].GHallmark: the number of non-synonymous mutations in cancer hallmark genes (GCH), according to the list described by Qing et al. and Hanahan and Weinberg [21,22].GRepRep: the number of non-synonymous mutations in genes involved in the replication or repair processes (GRepRep), listed in SF Appendix A.GCDH1path: the number of germline mutations with high functional impact in genes of the CDH1 pathway (KEEG: N258_257_61) [23].

A schematic flow chart illustration of the DryLab procedures [24] applied to this study is presented below (Figure 1).

## 3. Results

### 3.1. Mutations’ Impacts and Consequences

The impacts of the GC patients’ mutations and the consequences of the high-impact ones are presented in Figure 2A,B. Most mutations were classified as low or moderate. Among the high-impact mutations, the most frequent consequences were frameshift variants.

### 3.2. Number of Mutations

The total number of mutations in both the gastric cancer and non-cancer groups, as well as the number of selected mutations in each gene subset, are presented and compared.

A significant adj. *p* value was reached for the GMB as the total number of mutations in the GC group was much higher than that of the non-cancer group (adj. *p*: 0.008).

Regarding the selected types of mutations analyzed for each subset of genes, significant differences were found for GFI (adj. *p*: 0.000), GMed (adj. *p*: 0.000), and GHallmark (adj. *p*: 0.008), with higher numbers of mutations in the GC group.

Considering the GRepRep subset, the GC groups presented many more mutations than the control group; a statistical impact was not reached due to a reduced number of mutations in both groups and the limited number of included samples.

Regarding mutations in the CDH1 gene and genes involved in the CDH1 pathway, our results showed a scarcity of mutations in both groups (none of them in the CDH1 gene), as presented in Figure 3.

### 3.3. Most Mutated Genes and Correlations to Clinical Characteristics

The most frequently mutated genes are presented in the following plots, together with the clinical data and the impact of each mutation, favoring an integrated view of mutations frequencies, distributions, types, and their relationship with clinical features (Figure 4).

The MUC gene families were among the most prevalent. Regarding high-impact mutations, the LILRB1 gene was found to be a promising target for validation and deep investigation. The frequencies of these mutations were null in every consulted data bank (GNOMAD, ABraOM, ExAC, UK10K) and presented mutations in 31 cases of GC and 13 non-cancer controls, arising as a putative cancer risk gene for this population.

Note that the CDH1 and CTNNA1 genes do not appear in this list. There were no pathogenic variants currently related to HGC in these genes among the almost 70,000 mutations that were found in this series of 95 young GC patients.

### 3.4. Most Common Variants and Clinical Features

After applying the relevant filters, we generated a plot for each reported subset (again in an integrated view with clinical data) regarding the most frequent gene variants in each subset of genes, such as GHFI subset (Figure 5); GMed subset (Figure 6); GHallmark subset (Figure 7); GRepRep subset (Figure 8); GCDH1path subset (Figure 9).

Although not reaching statistical significance, a table presenting the adj. *p* value for every gene subset correlation with each set of clinical data is presented below (Table 1).

## 4. Discussion

GC in young people is a challenging medical situation due to the usual aggressiveness of such cases. There are many tragic consequences, such as severe physical restrictions and even death, for individuals just beginning their productive life and constructing their families and histories [25,26].

Moreover, the unsafe feelings that affect a patient’s relatives and health care providers impact the quality of life of all involved [7,27,28].

A few molecular germline mutations are currently recognized as causal events related to these cases [29]. There are formal recommendations for carriers and their relatives in these situations [30,31]. Nonetheless, many young GC patients do not have such mutations, as is the case in the 95 GC patients under 50 years old discussed in this study. Consequently, no specific measures are implemented to identify relatives at risk or look for cancer-risk-reduction strategies.

Most investigations aiming to identify germline variants implicated in cancer risk are already focused on described genes or variations related to each specific disease or, in the case of non-target exploration, are mainly based on associations of genetic variations and cancer occurrence. Genome-wide association studies (GWAS) are the most robust strategies usually applied to such discoveries [32,33,34].

The main principle of GWAS is the investigation of a large number of cases and controls (thousands) to reliably identify variants that are more frequently present among cases than among controls [35,36]. This principle is based on the hypothesis that specific variations implied in genetic cancer risk are recurrently present among cases and absent among controls.

Although favoring many vital discoveries in cancers and other diseases, GWAS do not cover all the potential genetic risk factors affecting cancer onset. Looking for recurrent genetic variations specific to an investigated cancer may underestimate the importance of some rare genetic variations [37].

Every cancer-related mutation that modifies a relevant protein for the cancer process can favor carcinogenesis, even if it rarely occurs. Accordingly, a genetic variant rarely found among cancer groups could avoid identification as a relevant risk factor for those cancers in GWAS and may not be considered “harmful.”

Nonetheless, it must be highlighted that cancer occurrence depends on many additional circumstances.

Germline mutations increase the risk of cancer but are not usually enough to cause adult cancers, including GC, without additional somatic mutations affecting the target tissue [38,39].

The related consequences of germline gene mutations for embryonic development phases might be disastrous if stem cells avoid differentiation, increase divisions or permit highly deleterious errors to be transmitted to the daughter stem cells, thus affecting body development [40].

Consequently, most germline cancer mutations occur in only one tumor suppressor gene allele, preserving tissues and organ development. The second hit, a somatic event reaching the normal allele, might finally complete that cancer driver event [39].

Even if homozygotes germline mutations occur, the affected gene position in such a situation is not usually strong enough to cause cancer. These mutations are either insufficient to cause events or are counteracted by redundant controls or compensatory mechanisms. Nonetheless, they might represent a significant additional risk factor for cancer development [41].

Reported exomes from clinical cases of HGC have demonstrated plenty of germline functional mutations in genes not recognized as causal germline GC mutations [42,43]. The results presented by our team are also teeming with conceptually non-relevant germline events that affect cancer genes. Even though they are able to contribute to cancer risk, these mutations do not require formal procedures or special attention.

The hypothesis that patients with inherited mutations in cancer-related genes might have an advantage in the oncogenic process was supported by this study. Accordingly, having more than one germ line functional mutation potentially increases the chance of somatic second-hit events. In other words, starting the cancer ride with many potential targets for a second somatic event might enhance cancer risk and favor the early onset of cancer.

Additionally, mutations in genes involved in the replication and DNA repair processes, if present, could favor the accumulation of additional non-repaired mutations, thus increasing cancer risk [44].

Moreover, mutations with a high functional impact, although not necessarily linked to a specific disease, may also increase the risk for diverse diseases, including cancer [45].

There are also genes recognized to be involved in medical diseases that, if mutated, could favor the break of healthy homeostasis and eventually contribute to cancer onset [46].

Furthermore, genes involved in cancer hallmarks are hot spots for every cancer investigation, so these genes were also checked to see whether being affected by germline mutations could increase cancer risk [21].

Regarding HGCs (mainly represented by DHGC), in the absence of CDH1 pathogenic mutations, investigating genes included in the CDH-1 pathway might enable the identification of alternative mutations worth exploring [12,47,48].

In this study, we performed a deep investigation that included the complete exome sequencing of 95 early-onset GC patients under 50 years old and a control group of 94 non-cancer Amerindians. We found robust evidence that a higher number of germline mutations seems to be related to a higher risk of GC and that mutations in subsets of genes carrying functional consequences are putatively involved in cancer processes.

A potential criticism of the proposed hypothesis is whether the reported mutations are germline. Since blood samples were collected in adulthood, some of the mutations found might be somatic blood mutations that are unrelated to the carcinogenic process. Considering this argument, the number of expected mutations should be much higher among older than younger people, since the rate of somatic mutations increases with aging. Additionally, regarding the non-cancer control group, blood samples were similarly collected in adulthood, validating these comparisons [49].

Additionally, reporting data from Amerindians and comparisons of these data with patients carrying part of this genetic ancestry may also favor relevant discoveries. Most reported cancer variants come from data banks lacking Amerindian samples. Accordingly, a variant specific to that ancestry might never be reported, regardless of its importance as a putative cancer risk variant [50,51].

This seems to be the case for LILRB1 gene variants, which are absent in the world’s cancer datasets but were present in high frequencies among these GC patients and Amerindians. These variants, if validated, may represent essential cancer variants specific to the Amerindian ancestry’s contribution [50].

An alternative interpretation might consider these variants related to the population but not to GC risk. This hypothesis would be true if the variants’ frequencies were more common among Amerindians than among the admixed GC group that carries no more than 30% of the Amerindian genetic background. However, this interpretation was incorrect, since the variants were found to be about three times more common in the GC group.

The proposed hypothesis implicates that these variants are involved in HGC risk and, if present in the admixed population, increase the risk for GC together with other genetic and environmental risk factors. Although present among Amerindians, their GC incidence is unknown. Additionally, they do not seem exposed to many environmental cancer risk factors.

These findings open an avenue to investigate additional variants and, mainly, to consider a low-cost blood test to search for such variants to identify people at risk or even asymptomatic GC patients, thus favoring earlier diagnosis and better outcomes.

Regarding GRepRep mutations, the GC group also presented a much higher number of relevant mutations than the control group, though without reaching statistical significance due to the small number of mutations in both groups.

We also demonstrated that a higher number of GRepRep mutations was related to a higher GMB. Nonetheless, a causal relationship cannot be proved by using just these data, since the higher number of GRepRep mutations may have only been a consequence of the increased GMB [44,52].

Additionally, this study found many more mutations in genes with a high functional impact on those related to medical diseases and in cancer hallmark genes among cancer patients than in the non-cancer group. Although these genes are putative risk factors for developing diseases, including cancer, they might also be representative of the whole scenario of the higher GMB in the cancer group.

Conversely, the number of mutations in the CDH1 pathway was found to be irrelevant, enhancing the hypothesis of existing alternative germline contributions to be explored.

Nonetheless, these findings do not correlate with the clinical data. The small number of cancer patients in each clinical situation may justify these negative results, and this correlation needs to be explored in a more extensive series of cases.

Performing the proposed non-restricted exploration of genetic variants among young patients with GC revealed plenty of potential cancer-related variations not previously considered HGC risk factors. The reported germline mutations must be explored as promising biomarkers of GC risk, opening a new perspective for discovering and translating biomarkers to aid the clinical management of these aggressive neoplasias.

Additionally, if confirmed as germline risk factors for hereditary GC, these new findings may redefine epidemiological data, thus increasing the number of unrecognized cases of hereditary cancers.

## 5. Conclusions

Currently, the number of molecularly confirmed HGCs is minimal despite the number of patients that fulfill the clinical criteria for considering this diagnosis.

The analyses of the germline mutation burden strongly demonstrated that the group of young GC patients presented many more mutations than the non-cancer control group, suggesting a role for previously non-valorized germline mutations in HGC risk.

The number of mutations identified by exome sequencing from the blood of early-onset GC was much higher than that from individuals without a cancer diagnosis. In particular, the number of high functional impact mutations, including those affecting genes involved in medical diseases, cancer hallmark genes, and DNA replication and repair processes, were much higher, strengthening the hypothesis of the potential causal role of such mutations in hereditary cancers.

Conversely, classically related HGC mutations were not found and the number of mutations in genes in the CDH1 pathway was not found to be relevant among these young GC patients, reinforcing the hypothesis of existing alternative germline contributions favoring the early onset of GC.

Identifying non-reported GC variants, potentially originating from under-studied populations, may pave the way for additional discoveries and the translation of this knowledge to clinical interventions for GC management.

The proposed new approaches may reduce the discrepancy between clinically suspected and molecularly proven hereditary GC, as well as shedding light on similar inconsistencies among other cancer types. Additionally, the results of this study may support the development of new blood tests for evaluating cancer risk in clinical practice, helping physicians make decisions about strategies for surveillance and risk-reduction interventions.

## Figures and Tables

**Figure 1 cancers-14-06125-f001:**
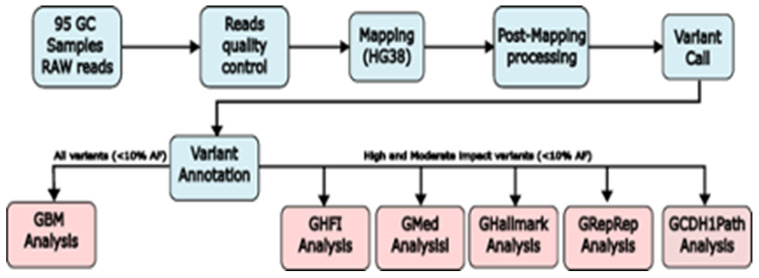
Schematic flow chart of the procedures applied.

**Figure 2 cancers-14-06125-f002:**
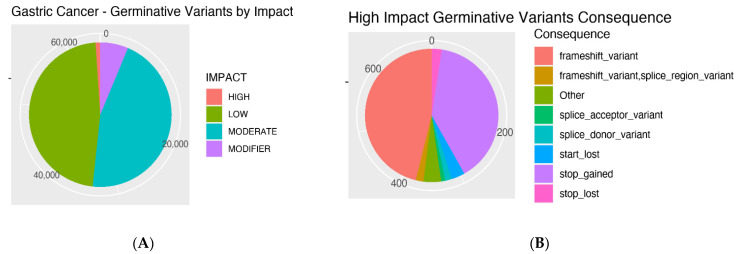
Gastric cancer germinative variants by impact: high, low, moderate and modifier (**A**); Gastric cancer high-impact germinative variants consequence: frameshift, frameshift/splice region, splice acceptor, splice donor, start lost, stop gained, stop lost and others (**B**).

**Figure 3 cancers-14-06125-f003:**
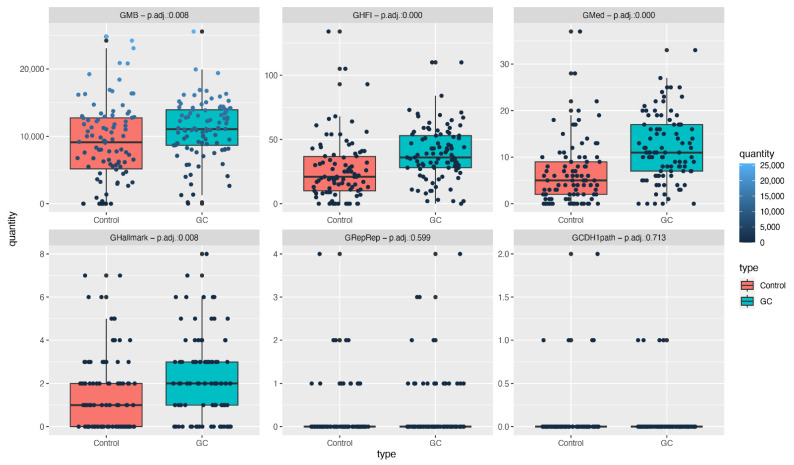
Quantity of mutations in gastric cancer (blue) and non-cancer (red) groups. There were more mutations in the GC group, reaching statistical significance for the total number of mutations (GMB) in the GHI, GMed and GHallmark gene subsets. No statistical differences occurred in the GRepRep or GCDH1path subsets.

**Figure 4 cancers-14-06125-f004:**
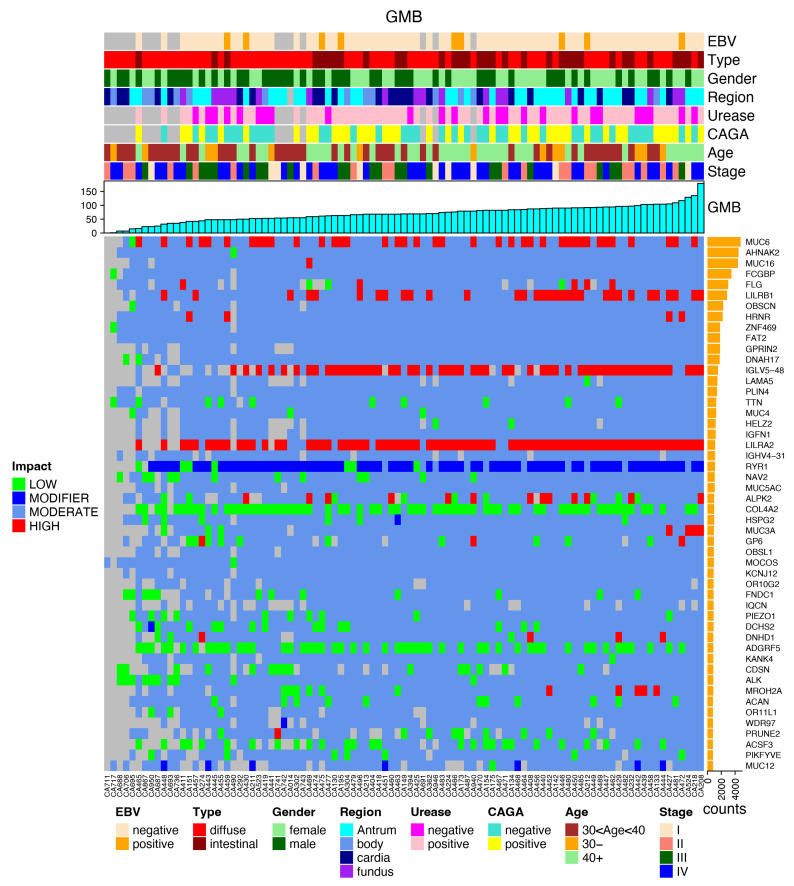
Most mutated genes in the GC group, the clinical data (EBV status, histological type, gender, region of the stomach, urease status, CAGA status, age group and stage) and the germline mutation burden of each case are correlated to the mutated genes and to the impact of mutations (low, modifier, moderate or high).

**Figure 5 cancers-14-06125-f005:**
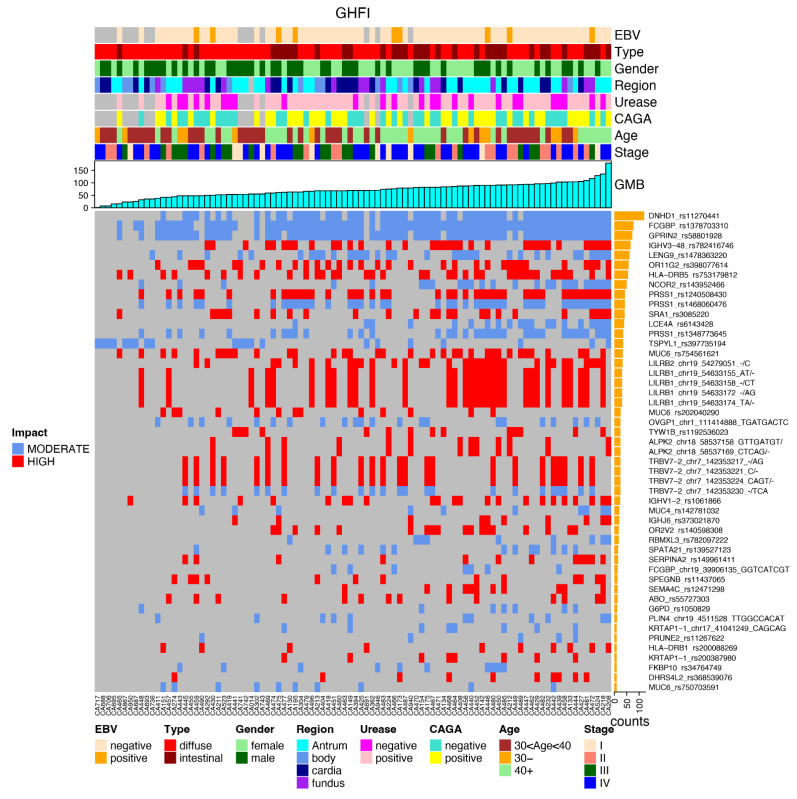
Correlation of most prevalent variants in the GHFI gene’s subset among young patients with gastric cancer and their distribution according to clinical features (EBV status, histological type, gender, region of the stomach, urease status, CAGA status, age group and stage), GMB and mutation impact (moderate or high).

**Figure 6 cancers-14-06125-f006:**
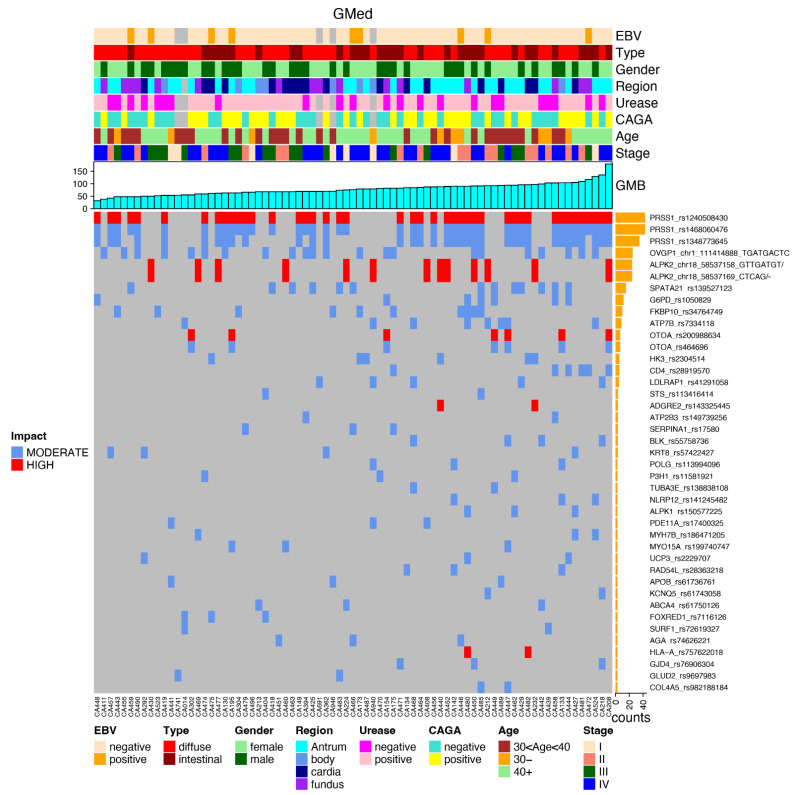
Correlation of most prevalent variants in the GMed gene’s subset among young patients with gastric cancer and their distribution according to clinical features (EBV status, histological type, gender, region of the stomach, urease status, CAGA status, age group and stage), GMB and mutation impact (moderate or high).

**Figure 7 cancers-14-06125-f007:**
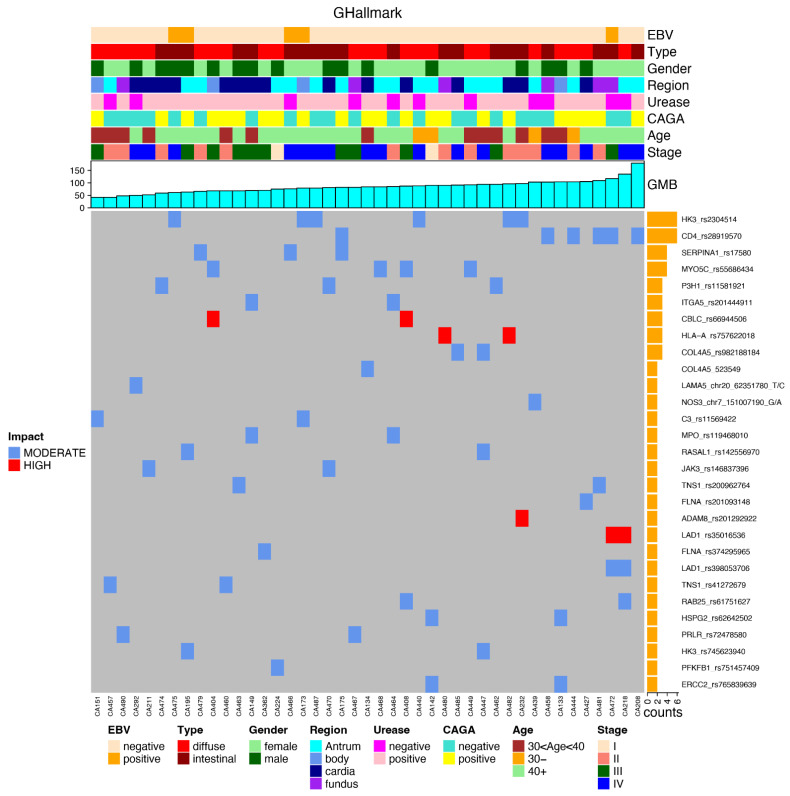
Correlation of most prevalent variants in the GHallmark gene’s subset among young patients with gastric cancer and their distribution according to clinical features (EBV status, histological type, gender, region of the stomach, urease status, CAGA status, age group and stage), GMB and mutation impact (moderate or high).

**Figure 8 cancers-14-06125-f008:**
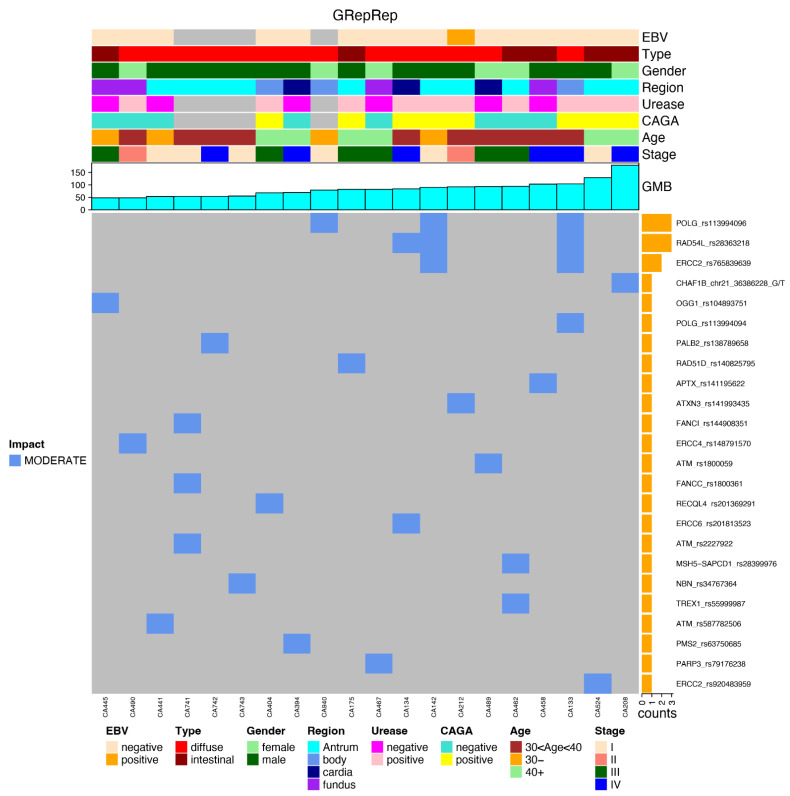
Correlation of most prevalent variants in the GRepRep gene’s subset among young patients with gastric cancer and their distribution according to clinical features (EBV status, histological type, gender, region of the stomach, urease status, CAGA status, age group and stage), GMB and mutation impact (moderate or high).

**Figure 9 cancers-14-06125-f009:**
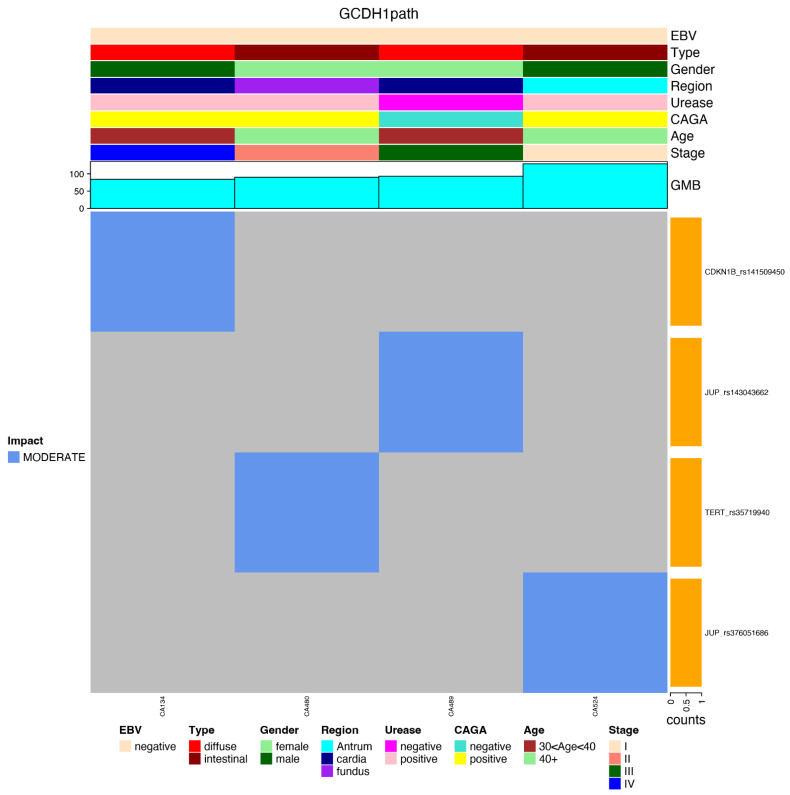
Correlation of most prevalent variants in the GCDH1path gene’s subset among young patients with gastric cancer and their distribution according to clinical features (EBV status, histological type, gender, region of the stomach, urease status, CAGA status, age group and stage), GMB and mutation impact (moderate or high).

**Table 1 cancers-14-06125-t001:** Statistical analysis of subsets by clinical data; FDR was used to adjust *p*-values.

		GMB	GHFI	GMed	GHallmark	GRepRep	GCDH1path
**Gender**	adj.p	0.12640	0.12640	0.47426	0.99146	0.12640	0.99146
**H. type**	adj.p	0.01328	0.09556	0.57447	0.10885	0.57447	0.57447
**Stage**	adj.p	0.89760	0.8976	0.89760	0.89760	0.03529	0.89760
**Age**	adj.p	0.32287	0.44947	0.7559	0.05163	0.51499	0.75590
**PCR.urease**	adj.p	0.88930	0.88930	0.88930	0.88930	0.88930	0.88930
**CAGA**	adj.p	0.99433	0.99433	0.99433	0.99433	0.99433	0.99433
**EBV**	adj.p	0.87695	0.87695	0.90029	0.87695	0.87695	0.87695
**Region**	adj.p	0.24500	0.16219	0.24500	0.24500	0.6612	0.66120
**EBV_urease**	adj.p	0.93363	0.93363	0.93363	0.93363	0.93363	0.93363
**EBV_CAGA**	adj.p	0.99597	0.99597	0.99597	0.99597	0.99597	0.99597

## Data Availability

The data presented in this study are available on request from the corresponding author.

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
