# Peer review of "Incidence of Hereditary Gastric Cancer May Be Much Higher than Reported"

_cancers, 2022, doi:10.3390/cancers14246125_

Round 1

Reviewer 1 Report

Dear Authors

Thank you very much for your manuscript submission. Please do revise your manuscript as below:

1. Please do add an abstract to manuscript.

2. It is recommended to use "DRY LAB" term for Bioinformatic Analysis; In this regard, please do read and add the following paper to References section of the manuscript:

DNA microarray technology and bioinformatic web services. Acta Microbiol Immunol Hung. 2019 Mar 1;66(1):19-30. doi: 10.1556/030.65.2018.028. Epub 2018 Jul 16. PMID: 30010394.

3. It is recommended to add a schematic flow chart to show the procedures applied in this study. Showing the Material and Methods and the Results section in this flow chart may be very helpful for the readers.

4. The legends pertaining to figures are not self-explanatory. Please do revise all of them.

5. It is recommended to bold the results pertaining to statistical analyses. It is important to interpret the statistical analyses clearly. Interpretation of significant correlations between different items associated with gender, age range, the related genes, etc. may be very useful.

Author Response

Dear Reviewer

Thanks for the relevant suggestions.

Every suggestion was accepted and included in the new version of the manuscript, as described below.

1- Please do add an abstract to manuscript.

   A: The abstract was included

2- It is recommended to use "DRY LAB" term for Bioinformatic Analysis; In this regard, please do read and add the following paper to References section of the manuscript:

DNA microarray technology and bioinformatic web services. Acta Microbiol Immunol Hung. 2019 Mar 1;66(1):19-30. doi: 10.1556/030.65.2018.028. Epub 2018 Jul 16. PMID: 30010394.

A: The DRY LAB term was added, and the suggested reference was included

3- It is recommended to add a schematic flow chart to show the procedures applied in this study. Showing the Material and Methods and the Results section in this flow chart may be very helpful for the readers.

A: The flow chart was included.

4- The legends pertaining to figures are not self-explanatory. Please do revise all of them

A: The legends were revised and modified

5 - It is recommended to bold the results pertaining to statistical analyses. It is important to interpret the statistical analyses clearly. Interpretation of significant correlations between different items associated with gender, age range, the related genes, etc. may be very useful.

A: None of the cited items have reached statistical significance.

Reviewer 2 Report

The present article, entitled,"Incidence of Hereditary Gastric Cancer May be Much Higher Than Reported" by Santos et.al., should be uploaded again.

1] The very first line of introduction is incomplete. It could be because authors uploaded a wrong/raw/incomplete file.

2] Abstract, keywords, abbreviations etc were not found in the article. 

3]Title of this article, Introduction and discussion sections require extensive editing of English language and style.

4] Style of references require major improvement. They were presented numerically but the body of article applied citations in different format.

5] Arrange and edit the article properly (in an acceptable format). Presentation is unfortunately not unto the par. eg., the method section involves a heading in Line #79- "Extraction of the DNA and Preparation of the Exome Library" . It took a second to realize that its a subsection!

This reviewer recommends extensive editing and advices authors to incorporate above suggestions and resubmit this manuscript for reconsideration in MDPI journal 'Cancers'.  

Author Response

Dear Reviewer

Thanks for the comments.

Every suggestion was accepted, and the corrections were included in the new version of the manuscript as described below.

1] The very first line of introduction is incomplete. It could be because authors uploaded a wrong/raw/incomplete file.

A: The correct file was uploaded.

2] Abstract, keywords, abbreviations etc were not found in the article. 

A: They were included in the current version.

3]Title of this article, Introduction and discussion sections require extensive editing of English language and style.

A: Professional English editing was provided ( MDPI English-55126).

4] Style of references require major improvement. They were presented numerically but the body of article applied citations in different format.

A: the references were fixed and uniformized. 

5] Arrange and edit the article properly (in an acceptable format). Presentation is unfortunately not unto the par. eg., the method section involves a heading in Line #79- "Extraction of the DNA and Preparation of the Exome Library" . It took a second to realize that its a subsection!

A: The article was edited. 

Round 2

Reviewer 2 Report

The present article, entitled, "Incidence of Hereditary Gastric Cancer May be Much Higher Than Reported", written by Assumpção et. al., has been improved efficiently as per the previous report's suggestions.  The title of the article could be improved (not impressed with the choice of the title)! But I leave that decision upto the editors and authors in this case.